# The Fairness Hierarchy: A viewpoint from causal inference

**ChengBo Zhang** [1]  **Zhen Yao** [1]  **Hao Pang** [1]  **Changcheng Li** [1]

## Abstract

Fairness in machine learning prediction has attracted growing attention in recent years. In this article, we propose a causal–inference–based framework for fair prediction, defined through path-specific counterfactual interventions. Instead of imposing fairness via constraints on predictive objectives or model parameters, our approach specifies fairness directly at the level of counterfactual prediction semantics. Given a well-specified causal graph, we construct a predictive distribution for the outcome $Y$ using a structural causal model and generate counterfactual predictions by selectively intervening on causal paths emanating from sensitive attributes. By allowing or blocking the propagation of sensitive information along designated paths, possibly involving multiple sensitive sources, our framework induces a hierarchy of interpretable fairness notions, generalizing standard path-specific causal semantics. Our empirical experiments demonstrate how different fairness levels can be instantiated and compared in practice.

## 1. Introduction

With the rapid development and widespread deployment of machine learning systems, fair prediction has emerged as a central concern in both research and practice. A large body of existing work adopts a distributional perspective on fairness (Madras et al., 2018; Cotter et al., 2019; Agarwal et al., 2018; Donini et al., 2018; Feldman et al., 2015), imposing constraints on the relationship (such as statistical independence) between a sensitive attribute $A$ and the prediction $\hat{Y}$. One prominent example is *demographic parity* (Zemel et al., 2013), which enforces that the distribution of predictions remains invariant across different sensitive groups. In addition

---

[1]School of Mathematical Sciences, Dalian University of Technology, Dalian, China. Correspondence to: Changcheng Li <lichangcheng@dlut.edu.cn>.

*Proceedings of the 43rd International Conference on Machine Learning*, Seoul, South Korea. PMLR 306, 2026. Copyright 2026 by the author(s).

to demographic parity, Hardt et al. (2016) introduced the notions of *equalized odds* and *equality of opportunity*, which require the prediction $\hat{Y}$ to be conditionally independent of $A$ given the true outcome $Y$. For a comprehensive overview of statistical notions of fairness and related developments, we refer the reader to (Barocas et al., 2023; He & Li, 2025).

In contrast to the distributional notion of fairness, Kusner et al. (2017) adopted a causal perspective on fairness by formulating fair prediction through counterfactual reasoning, treating the counterfactual outcome itself as the object of prediction. By leveraging the explicit semantics of structural causal models, this causal viewpoint grounds fairness in the underlying data-generating mechanism, and enables more fine-grained characterizations of fairness based on how sensitive attributes causally influence predictions. Within this causal framework, Kilbertus et al. (2017) proposed to intervene on proxy variables lying on causal paths from the sensitive attribute to the outcome, with the intuition that removing the influence of the sensitive attribute on these mediators prevents indirect discrimination. Under the counterfactual semantics, such interventions render the proxy variables invariant to changes in the sensitive attribute, thereby blocking certain causal effects. However, this approach does not address direct causal effects from the sensitive attribute to the outcome, and the choice of baseline values for intervened proxy variables lacks a clear causal interpretation.

In parallel with causal formulations that define fair prediction directly via counterfactual reasoning, another line of work enforces fairness by incorporating counterfactual considerations as constraints during model training (Nabi & Shpitser, 2018; Garg et al., 2019; Plečko & Bareinboim, 2024; Kher et al., 2025). These approaches construct counterfactual inputs by intervening on sensitive attributes or related features, and introduce additional regularization terms into the training objective to encourage invariant predictions across factual and counterfactual inputs. This constraint-based paradigm has been explored in a variety of settings, differing in how counterfactual inputs are generated and how invariance is enforced. While effective in practice, incorporating such regularization terms into the training objective inevitably alters the learned dynamics of the predictor. Consequently, the trained model may deviate systematically from the original data-generating mechanism, making the resulting predictions harder to interpret in causal terms.

Moreover, because fairness is enforced indirectly through regularization rather than explicit causal intervention, it is generally unclear whether such constraints may introduce new, unintended pathways through which the sensitive attribute influences the outcome. These issues raise concerns about the interpretability of the learned predictor and the potential emergence of new forms of discrimination that are not explicitly accounted for in the model design.

Building on the counterfactual prediction paradigm of Kusner et al. (2017), we propose a more refined notion of fair prediction that incorporates explicit normative distinctions at the level of causal pathways, allowing different levels of fairness in a fairness hierarchy. Rather than treating all causal influences of the sensitive attribute in a uniform manner, our framework recognizes that different pathways through which the sensitive attribute affects the outcome may carry distinct normative interpretations, and therefore calls for differentiated treatment in fair prediction. Our approach is inspired by the theory of path-specific causal effects (Avin et al., 2005), which provides a principled way to reason about causal influences transmitted along designated paths.

Existing path-specific counterfactual approaches typically regulate unfairness through interventions on a single sensitive attribute along designated impermissible paths (Chiappa, 2019; Wu et al., 2019; Chikahara et al., 2021; Yao et al., 2023; Zhu et al., 2023). However, in the presence of confounding, impermissible information reaching the outcome may not be attributable to a single sensitive variable alone. In such cases, intervening on a single sensitive attribute may be insufficient to flexibly regulate all impermissible causal influences.

Motivated by this observation, we develop a path-aware counterfactual framework that explicitly accounts for confounding by allowing different causal paths to be regulated through interventions at different locations. This enables multiple, path-dependent interventions to be specified within a single predictive construction, yielding a spectrum of normative fairness operations that selectively preserve admissible causal influences while blocking impermissible ones at the level of the underlying causal mechanism. From a practical perspective, this path-level formulation allows practitioners to specify which channels of sensitive information should be retained or suppressed when fairness concerns involve multiple sensitive or confounding variables. Moreover, since the intervention is defined at the level of counterfactual prediction semantics rather than through constraints on a particular loss function or architecture, the framework can be combined with flexible base predictors without modifying their internal training procedure.

Our main contributions can be summarized as follows:

- We propose a path-aware counterfactual framework for fair prediction that refines counterfactual fairness by introducing explicit normative distinctions at the level of causal pathways, allowing admissible and impermissible causal influences to be treated at different levels, thereby forming a fairness hierarchy.

- As a core component of the proposed framework, we introduce a multi-intervention formulation of counterfactual prediction that explicitly accounts for confounding structures in which confounders may exert impermissible influences on the prediction target. This formulation generalizes single-intervention approaches.

- We show that, within the proposed path-aware fairness hierarchy, a specific fairness level satisfies demographic parity. This characterization highlights the flexibility and interpretability of the proposed framework. And empirical results illustrate how such a fairness level and other levels can be instantiated in practice.

## 2. Selective-Intervention Framework

In this section, we formalize a framework that enables fairness control at the level of individual causal paths. We adopt the standard counterfactual notation and denote by $a$ the factual value of a sensitive variable, and by $a'$ a reference (baseline) value used for counterfactual comparison.

### 2.1. A Hierarchy of Path-Based Fairness Notions

**Notation and causal semantics.** We work under the standard structural causal model (SCM) framework. Each endogenous variable $V$ is generated by a structural equation $V := f_V(\mathrm{pa}(V), U_V)$, where $\mathrm{pa}(V)$ denotes the set of direct causal parents of $V$ in the underlying directed acyclic graph (DAG), and $U_V$ is an exogenous noise term. The joint observational distribution induced by the SCM is assumed to satisfy the Markov and faithfulness conditions with respect to the DAG, ensuring that the causal structure provides a well-defined semantic basis for counterfactual reasoning. For a comprehensive treatment of the SCM framework and its counterfactual semantics, we refer to Pearl (2009).

Throughout this work, we study fairness in prediction under a fully specified structural causal model (SCM) with a known DAG. We regard the SCM as an objective description of the data-generating process in the real world, and regard fairness as a normative constraint on how sensitive information is allowed to influence the prediction through this causal structure. Specifically, we adopt a path-based perspective: fairness is understood as the requirement that sensitive information may affect the prediction outcome $Y$ only through a designated set of admissible causal pathways, while its influence along impermissible pathways should be suppressed. This perspective is closely related to the notion

of path-specific effects (PSEs) in causal inference, which decompose causal reasoning according to designated causal pathways via nested counterfactuals.

In the causal inference literature, path-specific causal effects were originally introduced to characterize how the effect of a single intervention variable is transmitted to an outcome through different collections of causal paths (Avin et al., 2005). In this formulation, a designated treatment variable is intervened upon, and nested counterfactuals are used to selectively activate or deactivate its influence along specified paths. In our setting, we adopt the same counterfactual construction principle, but extend the scope of allowable interventions. Rather than restricting attention to a single intervention variable, we allow different causal paths to be regulated through interventions on different variables along those paths. This extension is motivated by the presence of confounding and complex causal structures in fairness applications, where suppressing impermissible information flow may require intervening at different locations depending on the path under consideration. The following definition builds on the nested counterfactual semantics for path- and edge-based interventions developed by Shpitser & Tchetgen Tchetgen (2016).

**Definition 2.1** (Admissible path-specific counterfactual semantics). Let $M$ be a structural causal model over endogenous variables $\mathcal{V}$. Let $\mathcal{S} \subseteq \mathcal{V}$ denote a set of sensitive variables, and let $Y \in \mathcal{V}$ be the outcome variable.

We assume that the set of directed causal paths from the sensitive set $\mathcal{S}$ to the outcome $Y$ is partitioned into admissible and impermissible sets, $\mathcal{P}_{\text{adm}}$ and $\mathcal{P}_{\text{imp}}$, respectively. For each impermissible path $\pi \in \mathcal{P}_{\text{imp}}$, a sensitive variable $S(\pi) \in \mathcal{S}$ is designated as the intervention target on that path.

Let $V \in \mathcal{V}$ be an endogenous variable with structural equation

$$V := f_V(\text{pa}(V), U_V).$$

For a given set of impermissible paths $\mathcal{P}_{\text{imp}}$, define $\text{pa}_{\text{imp}}(V)$ as the subset of parent inputs of $V$ corresponding to incoming edges that lie on impermissible paths and are downstream of the designated sensitive variable $S(\pi)$ on those paths. Let $\text{pa}_{\text{adm}}(V)$ denote the remaining parent inputs.

The path-specific counterfactual of $V$ is defined recursively in topological order by

$$
\begin{aligned}
&V(\text{imp} : \mathbf{s}', \text{adm} : \mathbf{s}) \\
&:= f_V\big(\text{pa}_{\text{imp}}(V)(\mathbf{s}'),\ \text{pa}_{\text{adm}}(V)(\mathbf{s}),\ U_V\big).
\end{aligned}
$$

where $\mathbf{s}$ and $\mathbf{s}'$ denote factual and reference values of the corresponding sensitive variables, respectively, and each parent variable appearing in the arguments of $f_V$ is itself

evaluated as a path-specific counterfactual under the same $(\text{imp} : \mathbf{s}', \text{adm} : \mathbf{s})$ assignment, yielding a well-defined nested counterfactual construction. The partition is defined at the level of parent inputs (edges), rather than parent variables, so that the same variable may contribute to both admissible and impermissible inputs through different paths.

*Remark* 2.2 (Interpretation of the framework). The formulation above provides a flexible mechanism for expressing normative fairness requirements at the level of causal pathways. By allowing admissible and impermissible paths to be specified explicitly, together with path-dependent intervention targets, the framework enables domain knowledge and normative considerations to be incorporated directly into the prediction construction. Importantly, admissibility is defined at the level of paths rather than variables, allowing the same variable to participate in different pathways with different normative interpretations.

*Remark* 2.3 (Consistency with Pearl-style counterfactual semantics). The admissible path-specific counterfactual semantics defined above are consistent with the standard counterfactual semantics of structural causal models in the sense of Pearl. In particular, for any choice of admissible and impermissible path sets, the mixed-input counterfactuals constructed through path-dependent interventions can be equivalently represented as nested counterfactuals under a well-defined SCM, and therefore do not introduce counterfactual quantities beyond the scope of the standard SCM framework.

**On identifiability.** Throughout this work, we operate under the assumption that the relevant confounding variables are observed or partially observed. When confounding is entirely unobserved, identifiability issues are unavoidable, as is well known in the classical causal inference literature (Pearl, 2009). Even under favorable observational regimes, path-specific counterfactual quantities are not guaranteed to be identifiable from the observational distribution. In particular, as shown by Avin et al. (2005), certain graphical structures, such as so-called kite configurations, can lead to non-identifiability of path-specific effects even when all endogenous variables except for independent exogenous disturbances are observed. Since our framework builds on the same nested counterfactual semantics, analogous identifiability challenges may arise under our path-dependent intervention specification.

We emphasize, however, counterfactual reasoning in fairness applications often proceeds by combining the observed data distribution with substantive modeling assumptions and domain knowledge which reflect human judgments about the data-generating process and enable meaningful counterfactual predictions beyond what is identifiable from distributional information alone. Our framework is compatible with such assumptions, and a systematic treatment of identifiabil-

ity and estimation under specific model classes is beyond the scope of this paper.

## 2.2. Hierarchical Fair Prediction

In this subsection, we specialize the path-specific counterfactual semantics to the case of a single sensitive attribute, and study how selective interventions along designated causal paths induce well-defined counterfactual outcomes. We first define a path-based counterfactual outcome that isolates the causal effect of the sensitive attribute along admissible pathways, and then illustrate the resulted nested counterfactual prediction through a concrete example. We further discuss several basic properties of the proposed construction.

**Definition 2.4** (Single-Sensitive $\mathcal{P}_{\mathrm{adm}}$-Fair Outcome Semantics). Let $M$ be a structural causal model with a single sensitive attribute $A$, and let $Y$ denote the outcome variable. Let $\mathcal{P}_{\mathrm{adm}}$ be a pre-specified set of admissible causal paths from $A$ to $Y$, and let $\mathcal{P}_{\mathrm{imp}}$ denote its complement.

Throughout, we denote by $a$ the factual value of $A$ and by $a'$ a reference (baseline) value used for counterfactual comparison.

The $\mathcal{P}_{\mathrm{adm}}$-fair outcome is defined as

$$Y^{\mathrm{fair}} := Y(\mathrm{adm}:a,\ \mathrm{imp}:a'),$$

where the sensitive attribute takes its factual value $a$ along admissible paths in $\mathcal{P}_{\mathrm{adm}}$, and is intervened to the reference value $a'$ along impermissible paths in $\mathcal{P}_{\mathrm{imp}}$. All intermediate variables are evaluated according to the induced path-specific counterfactual semantics.

In particular, if $a = a'$, then the path-specific counterfactual reduces to the factual outcome, so that $Y^{\mathrm{fair}} = Y$.

Definition 2.4 characterizes fairness at the level of causal mechanisms, rather than in terms of statistical associations or predictive performance. It imposes a normative constraint on how information carried by a sensitive attribute is permitted to influence the outcome through a fixed and given causal structure. Concretely, this is achieved by selectively intervening on the sensitive attribute along impermissible causal paths, while preserving its factual value along admissible ones. This construction does not evaluate whether the underlying causal model itself is fair; instead, it provides a principled semantics for defining counterfactual outcomes under user-specified fairness constraints, conditional on the assumed causal structure.

Figure 1 provides an intuitive illustration of the proposed fairness framework. The diagram organizes admissible path sets according to the partial order induced by set inclusion. Along any given chain of the poset, moving downward corresponds to progressively smaller admissible path sets, which can be interpreted as imposing increasingly stronger fairness constraints.

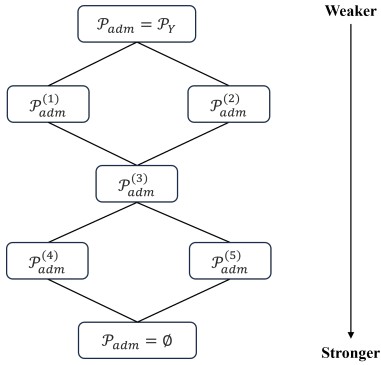

*Figure 1.* Hasse diagram of the partially ordered set of admissible path sets under the subset inclusion relation, where $\mathcal{P}_Y$ denotes the set of all directed causal paths from $\mathcal{S}$ to $Y$.

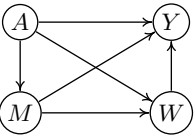

*Figure 2.* Causal model illustrating admissible and impermissible paths from the sensitive attribute $A$ to the prediction target $Y$.

As an illustrative example based on the causal mechanism depicted in Figure 2, we consider the causal paths $A \to M \to W \to Y$ and $A \to M \to Y$ as admissible, meaning that the influence of the sensitive attribute $A$ transmitted along these paths is regarded as legitimate. In contrast, the paths $A \to W \to Y$ and $A \to Y$ are treated as impermissible and should be removed from the prediction.

Under the structural causal model (SCM) semantics, the observational prediction of $Y$ takes the form

$$Y(a, M(a), W(a, M(a)), U_Y),$$

where each variable is evaluated according to its structural equation.

To obtain a fair counterfactual prediction that blocks the impermissible paths, we construct a path-specific counterfactual in which the contribution of $A$ is replaced by the reference value $a'$ only along impermissible paths. This yields the counterfactual predictor

$$Y^{\mathrm{fair}} = Y(a', M(a), W(a', M(a)), U_Y),$$

where the factual value $a$ is preserved along admissible paths, while the reference value $a'$ is substituted along impermissible paths. All counterfactual quantities are evaluated in a nested manner following the topological order of the SCM, as specified in Definition 2.1. This path-specific construction enables a finer-grained control of causal information flow, in which an intermediate variable such as $W$ may participate in both admissible and impermissible

pathways, while only the latter are selectively altered in the counterfactual outcome.

### 2.3. Selective Interventions with Sensitive Confounders

So far, our discussion has focused on selective interventions along directed causal paths originating from a single sensitive attribute $A$. This perspective is adequate when all ethically relevant information enters the system exclusively through $A$. In many applications, however, the sensitive attribute $A$ is itself influenced by upstream factors that also directly affect the outcome $Y$. We represent this situation by introducing a variable $Z$ that acts as a confounder of $A$ and $Y$, as illustrated in Figure 3. Although $Z$ does not lie on any directed causal path from $A$ to $Y$, it may itself carry sensitive information, such as background or historical characteristics, that is ethically relevant for prediction. As a consequence, causal paths emanating from $Z$ to $Y$ may also encode discriminatory effects and should not be automatically regarded as admissible. Whether the influence of $Z$ along specific paths is considered permissible or impermissible is inherently application-dependent and should be specified by the user or domain experts, in the same spirit as determining admissible and impermissible paths from $A$.

We now generalize the above formulation to settings in which multiple sensitive attributes are present. As before, fairness is assessed at the level of causal paths by distinguishing admissible from impermissible influences on the outcome. The essential difference from the single-sensitive case lies in the treatment of impermissible paths. When multiple sensitive attributes are involved, a single impermissible causal path may contain more than one sensitive variable, and intervening on different variables along the same path generally leads to different counterfactual outcomes. Consequently, specifying the set of impermissible paths alone is no longer sufficient. Instead, in addition to identifying impermissible paths, we must specify, for each such path, which sensitive attribute is subject to intervention. This leads to a generalized notion of fair outcome semantics, of which the single-sensitive definition is a special case.

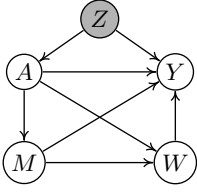

*Figure 3.* Observable confounder $Z$ positioned above the causal rectangle, affecting both $A$ and $Y$.

**Definition 2.5** (Multi-Sensitive $\mathcal{P}_{\mathrm{adm}}$-Fair Outcome Semantics)**.** Let $M$ be a structural causal model over endogenous variables $\mathcal{V}$. Let $\mathcal{S} \subseteq \mathcal{V}$ be a set of sensitive variables, and let $Y \in \mathcal{V}$ be the outcome variable.

We assume that the set of directed causal paths terminating at $Y$ is partitioned into admissible and impermissible sets, denoted by $\mathcal{P}_{\mathrm{adm}}$ and $\mathcal{P}_{\mathrm{imp}}$, respectively. For each impermissible path $\pi \in \mathcal{P}_{\mathrm{imp}}$, a sensitive variable $S(\pi) \in \mathcal{S}$ is designated as the intervention target on that path.

Let $\mathbf{s}$ denote the collection of factual values of the sensitive variables in $\mathcal{S}$, and let $\mathbf{s}'$ denote the corresponding collection of reference values. The $\mathcal{P}_{\mathrm{adm}}$-fair outcome is defined as

$$Y^{\mathrm{fair}} := Y(\mathrm{imp} : \mathbf{s}', \mathrm{adm} : \mathbf{s}),$$

where $Y(\mathrm{imp} : \mathbf{s}', \mathrm{adm} : \mathbf{s})$ is evaluated according to the admissible path-specific counterfactual semantics in Definition 2.1, using the path partition $(\mathcal{P}_{\mathrm{adm}}, \mathcal{P}_{\mathrm{imp}})$ and the path-wise intervention targets $\{S(\pi) : \pi \in \mathcal{P}_{\mathrm{imp}}\}$.

*Remark* 2.6 (Distributional Reference Values). In multi-sensitive settings, certain sensitive variables may act as confounders, for which a point-valued reference level is ambiguous or ill-defined. In such cases, the reference value $\mathbf{s}'$ may be interpreted distributionally, by intervening on the corresponding variable through sampling from a designated reference distribution.

*Remark* 2.7 (Operational Fair Prediction). The fair outcome $Y^{\mathrm{fair}}$ is defined at the level of causal counterfactual semantics. In practice, fair predictions are typically constructed using functionals of $Y^{\mathrm{fair}}$, such as conditional expectations or decision rules derived from them, when individual-level counterfactual realizations are inaccessible.

Definition 2.5 allows the user to flexibly specify, for each impermissible causal path, which sensitive variable is intervened upon. This path-dependent specification reflects the fact that in complex causal structures, different pathways may call for interventions at different locations, depending on how sensitive information propagates through the graph.

From an intuitive perspective, intervening on a sensitive variable closer to the outcome along an impermissible path effectively blocks the influence of all upstream sensitive information transmitted through that path. In contrast, intervening on a sensitive variable located earlier along the path may leave residual influence from downstream sensitive variables or exogenous disturbances that cannot be controlled through upstream intervention alone. This asymmetry highlights why a single, globally chosen sensitive variable is often insufficient to regulate all impermissible pathways in the presence of confounding or multiple sensitive attributes.

## 3. Causal and Distributional Notions of Fairness

The fairness notions introduced above are formulated at the level of causal mechanisms, rather than in terms of constraints on outcome distributions. In contrast, classical fairness criteria such as demographic parity are distributional in

nature, imposing statistical constraints on the marginal distribution of the outcome across sensitive groups. These two perspectives are conceptually distinct. Causal fairness regulates how sensitive information is allowed to influence the outcome through the underlying causal structure, whereas distributional fairness concerns properties of the induced outcome distribution. As a result, causal fairness does not, in general, imply distributional fairness, nor vice versa.

To relate our causal fairness notions to classical distributional ones, we recall the definition of demographic parity.

**Definition 3.1** (Demographic Parity). A predictor $\hat{Y}$ is said to satisfy demographic parity with respect to a sensitive attribute $A$ if

$$P(\hat{Y} \mid A = a) = P(\hat{Y} \mid A = a').$$

With this definition in place, it is natural to ask whether causal fairness mechanisms can recover demographic parity. The following proposition establishes a fundamental limitation in this regard. Even when path-specific counterfactual interventions are employed, regulating only a single sensitive attribute is, in general, insufficient to guarantee demographic parity at the distributional level.

**Proposition 3.2.** *Consider the single-sensitive path-specific fairness framework in which the sensitive set is given by $\mathcal{S} = \{A\}$. Let $\mathcal{P}_{\mathrm{adm}}$ be an arbitrary collection of admissible causal paths terminating at the outcome $Y$, and let $Y^{\mathrm{fair}}$ denote the fair outcome defined under the corresponding path-specific counterfactual semantics.*

*In general,*

$$Y^{\mathrm{fair}} \not\perp\!\!\!\perp A,$$

*and hence enforcing single-sensitive path-specific fairness does not, in general, imply demographic parity.*

*Remark* 3.3. The single-sensitive path-specific fairness framework considered above is defined at the counterfactual level and can be explicitly represented using structural causal models with shared exogenous variables. In this sense, it shares the same semantic capability as the level-3 framework in (Kusner et al., 2017), in that both approaches leverage the exogenous noise terms from the structural causal model to construct predictions. By further allowing selective interventions along designated causal paths, our framework strictly generalizes the path-based intervention formulation of (Chiappa, 2019), such that these prior approaches arise as special cases. Consequently, the limitation identified in Proposition 3.2 also applies to these approaches when interpreted under full counterfactual semantics.

The above limitation stems from intervening only on a single sensitive attribute. Although path-specific interventions on $A$ may successfully regulate the direct influence of $A$ along designated impermissible paths, they do not, in general, eliminate all impermissible causal pathways leading to the outcome. In particular, confounding variables $Z$ that causally influence both $A$ and $Y$ may themselves induce impermissible pathways to $Y$, even when the direct effect of $A$ has been appropriately constrained. Consequently, regulating $A$ alone is insufficient to ensure that all unfair causal influences on $Y$ have been removed.

We now show that this limitation can be overcome under the proposed multi-sensitive, path-dependent intervention framework. Specifically, we consider fixing the sensitive attribute $A$ at a reference value, while simultaneously treating all variables $Z$ that causally influence both $A$ and $Y$ as additional sensitive attributes. For each such variable $Z$, we replace its structural mechanism with a distributional intervention drawn from $P(Z \mid A = a')$. By jointly intervening on $A$ and these variables, the resulting counterfactual construction blocks all impermissible causal pathways—both direct and indirect— through which sensitive information can influence $Y$. Under this joint intervention scheme, we obtain a causal-level fairness mechanism that satisfies demographic parity at the distributional level. The following theorem formalizes this construction.

**Theorem 3.4** (Demographic parity via multi-sensitive distributional interventions). *Let $M$ be an SCM with an associated DAG $\mathcal{G}$ over endogenous variables. Let $A$ denote a sensitive attribute and $Y$ the outcome. Define*

$$\mathcal{Z} := \{Z \in \mathcal{V} \setminus \{A, Y\} : Z \rightsquigarrow A \text{ and } Z \rightsquigarrow Y\}.$$

*Here $U \rightsquigarrow V$ denotes the existence of a (possibly indirect) directed path from $U$ to $V$ in the DAG.*

*Let the sensitive set be $\mathcal{S} := \{A\} \cup \mathcal{Z}$, and set the admissible-path collection to be empty.*

*Consider the following multi-sensitive, path-dependent intervention scheme. The sensitive attribute $A$ is intervened to a fixed reference value $a'$. For each $Z \in \mathcal{Z}$, a distributional intervention is applied by drawing*

$$Z^{\star} \sim P(Z \mid A = a'),$$

*and, for each impermissible causal path to $Y$, the intervention is applied at the sensitive variable that is closest to $Y$ on that path.*

*Let $Y^{\mathrm{fair}}$ denote the resulting fair outcome defined by our multi-sensitive path-dependent counterfactual semantics under the above interventions. Then $Y^{\mathrm{fair}}$ satisfies demographic parity with respect to $A$, that is,*

$$P(Y^{\mathrm{fair}} = y \mid A = a) = P(Y^{\mathrm{fair}} = y \mid A = a'),$$
$$\forall a, a', y.$$

*Remark* 3.5 (Expectation-based counterfactual approximations). In practice, valid individual-level counterfactual realizations are often unavailable. A common alternative

is to work with conditional expectations over exogenous noise variables, yielding an expectation-based counterfactual approximation. This corresponds to the level-2 notion of counterfactual fairness in (Kusner et al., 2017). Under such expectation-based approximations, if all intermediate counterfactual variables are replaced by their conditional expectations while the reference group $A = a'$ is left unmodified, the resulting predictor generally fails to satisfy demographic parity. This failure arises from an asymmetry in the operational treatment of different sensitive groups. However, demographic parity can be restored by applying the same expectation-based correction procedure to the reference group $A = a'$ as well. That is, when both $A = a$ and $A = a'$ groups are processed through an identical expectation-based counterfactual pipeline, the resulting predictor satisfies demographic parity.

We use a simple running example to illustrate the preceding results. The causal structure of the example is depicted in Figure 4, where $Z$ acts as a confounder influencing both the sensitive attribute $A$ and the outcome $Y$. We specify the underlying SCM using the following linear-Gaussian equations:

$$A = \mathbb{I}\{Z + 0.5, U_A > 0\}, \quad Y = A + Z + U_Y.$$

Here, $U_A$ and $U_Y$ denote exogenous noise variables. We assume that $Z, U_A, U_Y \overset{\text{i.i.d.}}{\sim} \mathcal{N}(0,1)$. Throughout this example, we designate $A = 1$ as the reference value. Note that by symmetry, $P(A = 1) = P(A = 0) = 1/2$; however, since $A$ is generated via a thresholding mechanism on $Z$, the conditional distributions $Z \mid A = 1$ and $Z \mid A = 0$ differ in expectation (see Appendix A for details).

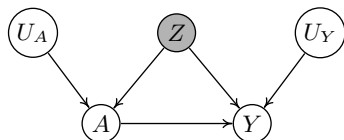

*Figure 4.* A simple example.

**Single-sensitive intervention.** Under the single-sensitive setting with $\mathcal{S} = \{A\}$ and $\mathcal{P}_{\text{adm}} = \varnothing$, Definition 2.4 yields $Y^{\text{fair}}(A) = 1 + Z + U_Y$ for $A \in \{0, 1\}$. Then, we have

$$\mathbb{E}[Y^{\text{fair}} \mid A = 1] \neq \mathbb{E}[Y^{\text{fair}} \mid A = 0],$$

which shows that intervening on $A$ alone does not, in general, guarantee demographic parity.

**Multi-sensitive intervention.** We now consider the multi-sensitive setting with $\mathcal{S} = \{A, Z\}$ and again $\mathcal{P}_{\text{adm}} = \varnothing$, fixing $A = 1$ as the reference group. For individuals with $A = 0$, we apply a distributional intervention $Z^\star \sim P(Z \mid A = 1)$. The resulting fair outcomes are $Y^{\text{fair}}(A = 0) =$

$1 + Z^\star + U_Y, Y^{\text{fair}}(A = 1) = 1 + Z + U_Y$. Since $Z^\star$ follows the same distribution as $Z \mid A = 1$ and is independent of $U_Y$, it follows immediately that

$$P(Y^{\text{fair}} \mid A = 0) = P(Y^{\text{fair}} \mid A = 1),$$

and hence demographic parity is satisfied.

Further conceptual discussions of our proposed framework are provided in appendix B.

# 4. Experiments

## 4.1. Synthetic Data

We first evaluate our method on synthetic data generated from a known structural causal model, shown in Figure 5. This controlled setting allows us to explicitly specify admissible and impermissible causal pathways and to verify the effectiveness of selective path interventions.

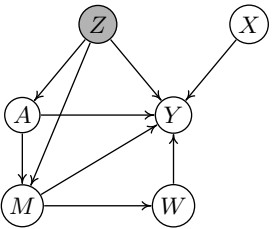

*Figure 5.* Causal graph used in the synthetic data experiments.

**Data-generation process.** The synthetic data are generated according to the following structural equations:

$$Z = U_Z, \quad X = U_X, \quad A = \mathbb{I}(Z + U_A > 0),$$
$$M = A + 0.5Z + U_M, \quad W = M + U_W,$$
$$Y = \mathbb{I}(0.5A + M + 0.5W + 0.5Z + 3X + U_Y - 1 > 0),$$

where all exogenous noise variables $\{U_Z, U_X, U_A, U_M, U_W, U_Y\}$ are mutually independent and follow standard normal distributions.

**Prediction under multiple fairness semantics.** In what follows, we focus on the semantic definition of fair outcomes. Given a specified fairness semantics, the corresponding prediction is always defined as the conditional expectation of the fair outcome given the sensitive variables $\mathcal{S}$ and the remaining covariates $X$, namely, $Y^{\text{pred}} = \mathbb{E}[Y^{\text{fair}} \mid \mathcal{S}, X]$. Throughout the simulation studies, we take $a = 1$ as the reference value for the sensitive attribute $A$ and we implement the prediction function using a $k$-nearest neighbors (kNN) regression model. We consider four prediction schemes induced by different intervention specifications.

Table 1 summarizes the four prediction settings considered in our experiments. The unconstrained prediction $\hat{Y}^{(0)}$ serves as a baseline with no fairness requirement. In both

*Table 1.* Summary of prediction settings under different fairness semantics.

| Prediction | Sensitive set $\mathcal{S}$ | Admissible paths $\mathcal{P}_{\mathrm{adm}}$ |
|---|---|---|
| $\hat{Y}^{(0)}$ | $\varnothing$ | $\varnothing$ |
| $\hat{Y}^{(1)}$ | $\{A\}$ | $\varnothing$ |
| $\hat{Y}^{(2)}$ | $\{A, Z\}$ | $\varnothing$ |
| $\hat{Y}^{(3)}$ | $\{A, Z\}$ | $\left\{ \begin{array}{l} Z \to M \to Y, \\ Z \to A \to M \to Y \end{array} \right\}$ |

$\hat{Y}^{(2)}$ and $\hat{Y}^{(3)}$, the sensitive variable $Z$ is intervened using a distributional intervention based on the reference distribution $P(Z \mid A = 1)$. In the setting $\hat{Y}^{(3)}$, for the admissible path $Z \to A \to M \to Y$, the intervention is applied at $A$ rather than at $Z$.

Taken together, Figures 6 and 7 illustrate an empirical pattern observed in our experiments under different fairness semantics. The unconstrained factual prediction $\hat{Y}^{(0)}$ achieves the highest overall accuracy, but exhibits a pronounced discrepancy in the predicted outcomes across the two sensitive groups. In contrast, the global multi-sensitive intervention $\hat{Y}^{(2)}$, which imposes the strongest constraint among the considered interventions, results in the lowest prediction accuracy in our experiments, while yielding nearly identical predictive distributions across groups. Overall, in our experiments, we empirically observe that stronger fairness constraints tend to be associated with lower prediction accuracy. Additional experimental details are deferred to Appendix D.

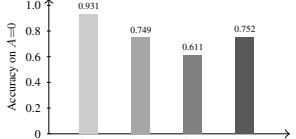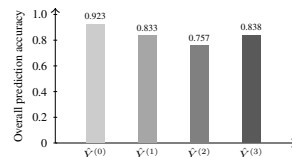

*Figure 6.* Prediction accuracy under different fairness semantics. Left: subgroup $A = 0$, where differences arise. Right: overall accuracy across both groups.

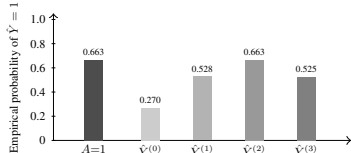

*Figure 7.* Empirical probability of predicting the positive outcome. The leftmost bar corresponds to the reference group $A = 1$, while the remaining bars show $\mathbb{P}(\hat{Y} = 1 \mid A = 0)$ under different fairness semantics.

**Additional analyses.** We further conduct three additional analyses in the synthetic setting to examine the empirical behavior of the proposed framework under graph misspecification, comparison with a Kusner-style predictor, and partial

observability of confounders. The data-generating mechanism is kept the same as in Section 4.1, and the results are summarized in Table 2. For graph misspecification, we use a misspecified working SCM obtained by adding the extra edge $Z \to W$, and then apply the same intervention strategy as $\hat{Y}^{(2)}$, where $(A, Z)$ are treated as sensitive attributes and the admissible path set is empty. For the Kusner-style comparison, we use residuals of descendants of the sensitive attribute $A$, together with non-descendant variables, as predictor inputs. For partial observability, we remove the confounder $Z$ from the causal graph used for prediction while keeping the underlying data-generating mechanism unchanged. This setting reflects the case where relevant confounding information is not explicitly represented in the working graph, but may still be partially transmitted through observed variables used for prediction.

Under mild graph misspecification, $\hat{Y}^{(\mathrm{mis})}$ yields group-wise positive prediction probabilities close to those of $\hat{Y}^{(2)}$, suggesting that the empirical behavior of the intervention is not substantially changed in this setting. The Kusner-style predictor $\hat{Y}^{(\mathrm{Kusner})}$ achieves a high overall accuracy of 0.840, but its group-wise positive prediction probabilities, 0.379 for $A = 0$ and 0.600 for $A = 1$, indicate a visible group-level gap. Finally, $\hat{Y}^{(\mathrm{partial})}$ attains a lower overall accuracy of 0.683, and its group-wise positive prediction probabilities, 0.762 for $A = 0$ and 0.663 for $A = 1$, show a clear group-level disparity. This illustrates that the empirical fairness behavior may be affected when relevant confounding information is only partially represented in the working causal graph.

### 4.2. COMPAS dataset

The COMPAS dataset (Correctional Offender Management Profiling for Alternative Sanctions) is a widely used benchmark for studying fairness in criminal justice risk assessment. It consists of pretrial records of defendants evaluated in Broward County, Florida, during 2013–2014, originally released by ProPublica (Angwin et al., 2016). In this study, we consider a binary prediction task of two-year recidivism. The outcome variable $Y$ indicates whether a defendant reoffended within two years. Following the dataset labels, race is treated as the primary sensitive attribute $A$, and we focus on the African-American and Caucasian groups, with the latter used as the reference group. Demographic information, including age and gender, is collected in the covariate set $Z$. Prior criminal history is represented by the mediator variable $M$. Figure 8 illustrates a simplified causal graph encoding the assumed relationships among these variables. In terms of predictive accuracy, the unconstrained baseline, corresponding to a factual prediction setting with no fairness intervention, achieves an accuracy of **0.657**. When the prediction rule is constrained to block the direct path from race $A$ to the outcome variable $Y$, while retaining the indirect

*Table 2.* Additional synthetic results under graph misspecification, Kusner-style prediction, and partial observability.

| Metric / Setting | $\hat{Y}^{(0)}$ | $\hat{Y}^{(1)}$ | $\hat{Y}^{(2)}$ | $\hat{Y}^{(3)}$ | $\hat{Y}^{\mathrm{mis}}$ | $\hat{Y}^{\mathrm{Kusner}}$ | $\hat{Y}^{\mathrm{partial}}$ |
|---|---|---|---|---|---|---|---|
| Overall accuracy | 0.923 | 0.833 | 0.757 | 0.838 | 0.747 | 0.840 | 0.683 |
| Accuracy on $A = 1$ | 0.931 | 0.749 | 0.661 | 0.611 | 0.620 | 0.846 | 0.531 |
| $\widehat{\mathbb{E}}[\hat{Y} \mid A = 1]$ | 0.270 | 0.528 | 0.663 | 0.525 | 0.654 | 0.379 | 0.762 |
| $\widehat{\mathbb{E}}[\hat{Y} \mid A = 0]$ | 0.663 | 0.663 | 0.663 | 0.663 | 0.663 | 0.600 | 0.663 |

Here $\hat{Y}^{\mathrm{mis}}$ is obtained under a mildly misspecified SCM. $\hat{Y}^{\mathrm{Kusner}}$ denotes the Kusner-style baseline using residualized descendants and non-descendants as predictors. $\hat{Y}^{\mathrm{partial}}$ denotes the setting where a relevant confounder is removed from the causal graph used for prediction while the data-generating mechanism remains unchanged. The conditional expectations report empirical group-wise positive prediction probabilities.

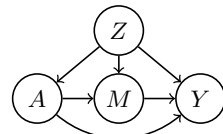

*Figure 8.* Causal graph for the COMPAS dataset.

path $A \to M \to Y$, the resulting accuracy remains nearly unchanged at **0.656**, indicating negligible performance loss in this experiment. In contrast, when demographic variables in $Z$ are further restricted from directly entering the prediction rule, such that the allowed information flow from $Z$ to $Y$ operates only through the path $Z \to A \to M \to Y$, the prediction accuracy decreases to **0.618**. This suggests that stricter constraints are associated with lower predictive accuracy in this experiment. Additional experimental details are provided in Appendix D.

## 5. Conclusion

This paper studies fairness from a causal-mechanism perspective and clarifies its relationship with classical distributional fairness criteria. We show that path-specific fairness based on a single sensitive attribute is, in general, insufficient to guarantee demographic parity, even under full counterfactual semantics.

To address this issue, we propose a multi-sensitive, path-dependent counterfactual framework that allows simultaneous interventions on multiple sensitive variables. By combining fixed-value and distributional interventions, the proposed semantics enables selective suppression of unfair causal influences while preserving admissible mechanisms. We establish that, under this framework, demographic parity can be achieved without requiring global interventions or identifiability of joint counterfactuals.

Several limitations of the present work suggest directions for future research. First, our framework relies on a specified structural causal model, and its conclusions may be affected by misspecification of the underlying causal structure. Second, some levels of the proposed fairness hierar-

chy require that relevant confounders are at least partially observed. Third, our theoretical analysis is primarily formulated at the population level and does not explicitly account for statistical estimation errors. An important extension is therefore to develop statistically efficient estimation procedures for multi-sensitive distributional interventions under finite-sample constraints and model misspecification. It is also of interest to integrate the proposed framework with representation learning, enabling scalable fair prediction in high-dimensional settings. Finally, exploring the relation between causal-level fairness and other distributional criteria beyond demographic parity is a promising direction.

## Acknowledgements

We thank the anonymous reviewers for their constructive comments. The corresponding author was supported by Xiaomi Foundation.

## Impact Statement

This paper presents work whose goal is to advance the field of Machine Learning. There are many potential societal consequences of our work, none which we feel must be specifically highlighted here.

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

# A. Proofs and Discussions of Propositions and Theorems

### Causal effect in Definition 2.4.

**Proposition A.1.** *Fix an admissible path set* $\mathcal{P}_{\mathrm{adm}}$. *For the protected group, defined as units with factual sensitive value* $A = a$, *the causal effect of* $A$ *on the* $\mathcal{P}_{\mathrm{adm}}$*-fair outcome satisfies*

$$\mathbb{E}[Y(a') - Y(a)] - \mathbb{E}[Y^{\mathrm{fair}}(a') - Y^{\mathrm{fair}}(a)] = \mathbb{E}[Y^{\mathrm{cf}}_{\mathcal{P}_{\mathrm{adm}}} - Y(a)]$$

*where* $Y^{\mathrm{cf}}_{\mathcal{P}_{\mathrm{adm}}} := Y(\mathrm{adm} : a, \ \mathrm{imp} : a')$. *In particular, the component of the total causal effect transmitted through impermissible paths is removed by construction.*

*Proof.*

$$\mathbb{E}[Y(a') - Y(a)] = \mathbb{E}[Y(a') - Y^{\mathrm{cf}}_{\mathcal{P}_{\mathrm{adm}}}] + \mathbb{E}[Y^{\mathrm{cf}}_{\mathcal{P}_{\mathrm{adm}}} - Y(a)],$$

By Definition 2.4, for units with factual $A = a$ we have

$$\mathbb{E}[Y^{\mathrm{fair}}(a') - Y^{\mathrm{fair}}(a)] = \mathbb{E}[Y(a') - Y^{\mathrm{cf}}_{\mathcal{P}_{\mathrm{adm}}}].$$

$\square$

Proposition A.1 establishes that the $\mathcal{P}_{\mathrm{adm}}$-fair outcome $Y^{\mathrm{fair}}$ removes, by construction, the causal influence of the sensitive attribute $A$ transmitted through impermissible paths, while preserving effects along admissible ones.

**Additional Discussion on Proposition 3.2** We emphasize that Proposition 3.2 does not assert that demographic parity is impossible under single-sensitive interventions whenever confounding variables are present. Rather, it states that regulating a single sensitive attribute is, in general, insufficient to guarantee demographic parity at the distributional level. In particular, there exist special data-generating distributions under which single-sensitive path-specific interventions may incidentally satisfy demographic parity, even in the presence of confounders.

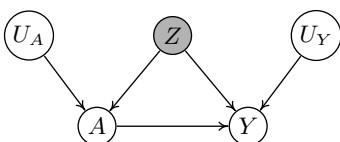

*Figure 9.* SCM used to illustrate exceptional cases

We consider the SCM shown in Figure 9. Let $U_A$, $Z$, and $U_Y$ be mutually independent exogenous variables, each following a standard normal distribution, i.e.,

$$U_A, Z, U_Y \overset{\mathrm{ind}}{\sim} \mathcal{N}(0, 1).$$

The sensitive attribute is generated via the threshold mechanism

$$A = \mathbb{I}\{U_A Z > 0\},$$

and the outcome variable is given by

$$Y = A + Z + U_Y.$$

We first characterize the posterior distribution of $Z$ given $A$. For any $z \neq 0$, note that

$$A = 1 \iff U_A z > 0 \iff \begin{cases} U_A > 0, & z > 0, \\ U_A < 0, & z < 0, \end{cases}$$

and hence, using $U_A \sim \mathcal{N}(0, 1)$,

$$P(A = 1 \mid Z = z) = \frac{1}{2}, \qquad P(A = 0 \mid Z = z) = \frac{1}{2}, \quad \forall z \neq 0.$$

Since $P(A = 1) = P(A = 0) = 1/2$ by symmetry, Bayes' rule yields

$$f_{Z|A=1}(z) = \frac{P(A = 1 \mid Z = z)f_Z(z)}{P(A = 1)} = f_Z(z), \qquad f_{Z|A=0}(z) = \frac{P(A = 0 \mid Z = z)f_Z(z)}{P(A = 0)} = f_Z(z),$$

that is, $Z \mid A = 1 \overset{d}{=} Z \mid A = 0 \overset{d}{=} Z$.

Under the single-sensitive setting with $\mathcal{S} = \{A\}$ and $\mathcal{P}_{\mathrm{adm}} = \varnothing$, Definition 2.4 implies that the fair outcome is obtained by fixing $A$ to its reference value 1, namely

$$Y^{\mathrm{fair}} = 1 + Z + U_Y.$$

Therefore,

$$P\big(Y^{\mathrm{fair}} \mid A = 0\big) = P(1 + Z + U_Y \mid A = 0) = P(1 + Z + U_Y \mid A = 1) = P\big(Y^{\mathrm{fair}} \mid A = 1\big),$$

which shows that demographic parity holds in this special case.

We present the detailed calculations omitted from the main text. Under the above specification, let $T := Z + 0.5\, U_A$, so that $A = \mathbb{I}\{T > 0\}$. Since $T$ is symmetrically distributed around zero, $A$ is a binary variable with $\mathbb{P}(A = 1) = \mathbb{P}(A = 0) = 1/2$.

Conditioned on $Z = z$, we have $T \mid Z = z \sim \mathcal{N}(z, 0.25)$, and hence

$$P(A = 1 \mid Z = z) = P(T > 0 \mid Z = z) = \Phi(2z),$$

where $\Phi$ denotes the standard normal distribution function. By Bayes' rule,

$$f_{Z|A=1}(z) = \frac{P(A = 1 \mid Z = z)f_Z(z)}{P(A = 1)} = 2\,\Phi(2z)\,\phi(z),$$

and similarly $f_{Z|A=0}(z) = 2\,\Phi(-2z)\,\phi(z)$. Here $\phi(\cdot)$ denotes the standard normal density. The conditional distributions above belong to the skew-normal family (Azzalini, 1985), whose mean admits a closed-form expression. In particular, we have

$$\mathbb{E}[Z \mid A = 1] = \sqrt{\frac{8}{5\pi}}, \qquad \mathbb{E}[Z \mid A = 0] = -\sqrt{\frac{8}{5\pi}}.$$

Under the single-sensitive setting with $\mathcal{S} = \{A\}$ and $\mathcal{P}_{\mathrm{adm}} = \varnothing$, Definition 2.4 yields

$$Y^{\mathrm{fair}} = 1 + Z + U_Y,$$

Taking conditional expectations and using $\mathbb{E}[U_Y \mid A] = 0$, we obtain

$$\mathbb{E}\big[Y^{\mathrm{fair}} \mid A = 1\big] = 1 + \mathbb{E}[Z \mid A = 1] = 1 + \sqrt{\frac{8}{5\pi}},$$

and

$$\mathbb{E}\big[Y^{\mathrm{fair}} \mid A = 0\big] = 1 + \mathbb{E}[Z \mid A = 0] = 1 - \sqrt{\frac{8}{5\pi}}.$$

We now turn to the proposed multi-sensitive semantics. Consider the sensitive set $\mathcal{S} = \{A, Z\}$ and again take $\mathcal{P}_{\mathrm{adm}} = \varnothing$. We fix the reference group to be $A = 1$. For individuals with $A = 0$, we apply a distributional intervention to the confounder by drawing

$$Z^\star \sim P(Z \mid A = 1).$$

For individuals with $A = 1$, no intervention on $Z$ is needed, since $Z \mid A = 1$ already follows the reference distribution. By Definition 2.5, the resulting fair outcomes under the multi-sensitive setting are given by

$$Y^{\mathrm{fair}}(A = 0) = 1 + Z^\star + U_Y, \qquad Y^{\mathrm{fair}}(A = 1) = 1 + Z + U_Y.$$

Since $Z^\star$ is drawn from the reference distribution, we have

$$P(Z^\star \mid A = 0) = P(Z^\star) = P(Z \mid A = 1).$$

Together with the independence between $Z^\star$ and $U_Y$, it follows that

$$P\big(Y^{\mathrm{fair}} \mid A = 0\big) = P\big(Y^{\mathrm{fair}} \mid A = 1\big).$$

**Proof of Theorem 3.4**

**Structural causal model and notation.**  Let

$$M = (\mathcal{U}, \mathcal{V}, P(\mathcal{U}), \{f_V\}_{V \in \mathcal{V}})$$

be a structural causal model (SCM) that is compatible with the directed acyclic graph $\mathcal{G}$. Here, $\mathcal{V}$ denotes the set of endogenous variables, and $\mathcal{U} = \{U_V : V \in \mathcal{V}\}$ denotes the collection of exogenous variables. The exogenous variables are jointly distributed according to $P(\mathcal{U})$, and each endogenous variable $V \in \mathcal{V}$ is generated by a structural equation of the form

$$V := f_V(\mathrm{pa}(V), U_V),$$

where $\mathrm{pa}(V)$ denotes the set of parents of $V$ in $\mathcal{G}$.

All random variables are defined on a common probability space $(\Omega, \mathcal{F}, \mathbb{P})$, and measurability is assumed throughout.

**A measurable representation of $Y$.**  Throughout this result, we consider the case where all directed paths from the sensitive information to the outcome are impermissible, i.e., $\mathcal{P}_{\mathrm{adm}} = \varnothing$. When multiple sensitive variables appear on the same directed path to $Y$, our path-dependent semantics replaces only the sensitive variable closest to $Y$, thereby avoiding any nested or ambiguous interpretation and ensuring that the construction is well-defined by a single forward simulation.

Under this semantics, there exists a measurable function $\phi$ such that

$$Y = \phi(A, Z, X, U),$$

where $(A, Z)$ denote the sensitive interface variables, $X$ collects all endogenous variables that have directed paths to $Y$ but do not lie on any directed path from $A$ or $Z$ to $Y$, and $U$ denotes the collection of exogenous disturbances that are independent of $(A, Z)$, and moreover $(A, Z) \perp (U, X)$.

We now justify this independence and the resulting representation. First, the exogenous disturbances associated with the sensitive interface variables $(A, Z)$ are not included in $U$, since their effects are already absorbed by the values of $(A, Z)$. Suppose, toward a contradiction, that $(A, Z)$ is not independent of $(U, X)$. Then there exists either an endogenous variable $X^\star \in X \subseteq \mathcal{V} \setminus \mathcal{S}$ or an exogenous disturbance $U_{X^\star} \in U$ and a sensitive node $Z^\star \in \mathcal{S}$ such that $X^\star \rightsquigarrow Z^\star$.

Since $X^\star \rightsquigarrow Z^\star$ with $Z^\star \in \mathcal{S}$ and $Z^\star \rightsquigarrow Y$, we have $X^\star \in \mathcal{S}$, hence $U_{X^\star} \notin U$ and $X^\star \notin X$, a contradiction.

Finally, consider any endogenous variable $M$ that lies on a directed path from $(A, Z)$ to $Y$. We show by induction on a topological ordering of the DAG that $M$ can be written as

$$M = \phi_M(A, Z, X_M, U_M, U),$$

where $X_M$ denotes the collection of endogenous variables that have directed paths to $M$ but do not lie on any directed path from $(A, Z)$ to $M$, $U_M$ is the exogenous disturbance associated with $M$, and $\phi_M$ is a measurable function and $U$ denotes a collection of exogenous disturbances, independent of $(A, Z)$, that are involved in the representation. Applying this argument recursively along directed paths to $Y$ yields

$$Y = \phi(A, Z, X, U)$$

for some measurable function $\phi$, which establishes the claimed representation.

**Conclusion.**  First consider the reference group $A = a'$. By construction, when $A = a'$, the distributional intervention on $Z$ coincides with its factual distribution, and hence

$$Y^{\mathrm{fair}}(A = a') = Y = \phi(a', Z, X, U).$$

Next consider $A = a$. Under the path-dependent semantics, the sensitive interface is replaced by the reference values, yielding

$$Y^{\mathrm{fair}}(A = a) = Y_{a \leftarrow a',\, Z \leftarrow Z^\star} = \phi(a', Z^\star, X, U),$$

where $Z^\star \overset{d}{=} Z \mid A = a'$.

Since $(A, Z) \perp (U, X)$, it follows that

$$\phi(a', Z^\star, X, U) \overset{d}{=} \phi(a', Z, X, U),$$

and therefore

$$P(Y^{\text{fair}} | A = a) = P(Y^{\text{fair}} | A = a').$$

## B. Additional Discussions on Fairness Notions

**Discussion on individual counterfactual fairness.**    Counterfactual fairness was first formally introduced by Kusner et al. (2017) as an individual-level fairness notion defined within a structural causal model. For reference, we state the definition below.

**Definition B.1** (Counterfactual fairness (Kusner et al., 2017))**.**  A predictor $\hat{Y}$ is said to be counterfactually fair if, under any context $X = x$ and $A = a$,

$$\mathbb{P}\left(\hat{Y}_{A \leftarrow a}(U) = y \mid X = x, A = a\right) = \mathbb{P}\left(\hat{Y}_{A \leftarrow a'}(U) = y \mid X = x, A = a\right),$$

$$\text{for all } y \text{ and for any attainable value } a' \text{ of } A.$$

In Kusner et al. (2017), the authors consider predictions constructed from a structural causal model by explicitly excluding the sensitive attribute and its descendants from the predictor. Under this construction, the resulting prediction satisfies the individual counterfactual fairness criterion in Definition B.1, as it depends only on exogenous noise variables that are invariant to interventions on the sensitive attribute.

Our framework operates at a finer granularity by explicitly modeling and intervening on path-level causal effects. As a result, the induced fair prediction does not necessarily satisfy individual counterfactual fairness at all levels of the admissible path hierarchy. However, in the special case where the sensitive set contains only a single attribute $A$ and the set of admissible paths is empty, i.e., $\mathcal{S} = \{A\}$ and $\mathcal{P}_{\text{adm}} = \varnothing$, our path-aware fair prediction coincides with the setting considered in Kusner et al. (2017) and satisfies the individual counterfactual fairness criterion.

Intuitively, this follows from the fact that our prediction is explicitly constructed using counterfactual outcomes under interventions, which naturally ensures invariance with respect to changes in the sensitive attribute when no causal path from $A$ to $Y$ is deemed admissible. We summarize this observation in the following proposition.

**Proposition B.2.** *Suppose that the sensitive set contains a single sensitive attribute $\mathcal{S} = \{A\}$, and that no causal path from $A$ to $Y$ is admissible $\mathcal{P}_{\text{adm}} = \varnothing$. Then, the resulting $\mathcal{P}_{\text{adm}}$-fair outcome $Y^{\text{fair}}$ is counterfactually fair in the sense of Definition B.1.*

*Proof.* When the admissible path set is empty, $\mathcal{P}_{\text{adm}} = \varnothing$, the $\mathcal{P}_{\text{adm}}$-fair outcome $Y^{\text{fair}}$ is constructed using counterfactual interventions that fix the sensitive attribute $A$. By Definition 2.4, this construction ensures that $Y^{\text{fair}}_{A \leftarrow a} = Y^{\text{fair}}_{A \leftarrow a'}$ for any attainable values $a, a'$ of $A$ for the same individual, and thus $Y^{\text{fair}}$ satisfies individual counterfactual fairness in Definition B.1.

$\square$

**Discussion on Equalized Odds.**    Another important distributional fairness notion is *Equalized Odds*. Unlike the definition introduced by Hardt et al. (2016) for binary classification settings, for ease of discussion we present below an equivalent formulation of Equalized Odds that allows the outcome variable $Y$ to be continuous.

**Definition B.3** (Equalized Odds)**.**  A predictor $\hat{Y}$ is said to satisfy *Equalized Odds* with respect to a sensitive attribute $A$ if

$$\mathbb{P}(\hat{Y} \mid Y = y, A = a) = \mathbb{P}(\hat{Y} \mid Y = y), \quad \forall y, \ \forall a, \ .$$

where the equality is understood in distribution.

In our framework, when the sensitive set is empty $\mathcal{S} = \varnothing$, no fairness constraint is imposed. In this case, the resulting fair outcome coincides with the factual outcome, namely $Y^{\text{fair}} = Y$. Consequently, $Y^{\text{fair}}$ trivially satisfies Equalized Odds, since conditioning on $Y$ renders $Y^{\text{fair}}$ degenerate and hence independent of any sensitive attribute.

Although this setting is trivial from a fairness perspective, it provides useful intuition. If the predictive model fully respects the true causal data-generating process, then the discrepancy between the model prediction and the ground truth should not systematically depend on sensitive attributes. In this sense, the structural causal model serves to encode the correct prediction mechanism.

In practice, however, predictions are not made directly using $Y^{\text{fair}}$, but rather through a regression function of the form

$$\hat{Y} = \mathbb{E}\big[Y^{\text{fair}} \mid \text{pa}(Y)\big],$$

where $\text{pa}(Y)$ denotes the set of direct parents of $Y$ in the causal graph. Assume that the structural equation for $Y$ follows an additive noise model, $Y = f(\text{pa}(Y)) + U_Y$, with noise term $U_Y$. Then we have

$$\mathbb{P}(\hat{Y} \mid Y, A) = \mathbb{P}(\hat{Y} - Y + Y \mid Y, A) = \mathbb{P}(U_Y + Y \mid Y, A) = \mathbb{P}(U_Y + Y \mid Y) = \mathbb{P}(\hat{Y} \mid Y),$$

which implies that $\hat{Y}$ satisfies Equalized Odds.

These observations highlight how different distributional fairness notions relate to the various causal fairness levels considered in our framework, clarifying both their connections and distinctions.

## C. Statistical and Computational Considerations

**Estimation error of distributional interventions.** Theorem 3.4 is a population-level result, where the reference intervention distribution is assumed to be known exactly. In practice, this distribution has to be estimated from data. When the sensitive attribute $A$ is binary, estimating $P(Z \mid A = a')$ reduces to estimating the distribution of $Z$ within the reference group $A = a'$, which can be done by the empirical conditional distribution. Specifically, a natural plug-in estimator is $\widehat{P}(Z \mid A = a')$, the empirical distribution of $Z$ based on samples with $A = a'$. Let $\widehat{Z}^\star \sim \widehat{P}(Z \mid A = a')$ denote the corresponding plug-in intervention variable. The exact demographic-parity conclusion in Theorem 3.4 relies on the identity $Z^\star \sim P(Z \mid A = a')$; after replacing this distribution by its estimate, one generally obtains an approximate version of demographic parity.

This approximation can be quantified using total variation distance. Let $d_{\text{TV}}(P, Q) = \sup_B |P(B) - Q(B)|$. Suppose that the fair prediction under the population intervention can be written as

$$Y^{\text{fair}} = \phi(a', Z^\star, X, U),$$

where $U$ denotes the exogenous randomness in the structural causal model and the prediction rule. The plug-in version is

$$\widehat{Y}^{\text{fair}} = \phi(a', \widehat{Z}^\star, X, U).$$

Since both variables are generated through the same structural and prediction map, the induced deviation satisfies

$$d_{\text{TV}}\Big(\mathcal{L}(\widehat{Y}^{\text{fair}} \mid A = a), \mathcal{L}(Y^{\text{fair}} \mid A = a')\Big) \leq d_{\text{TV}}\Big(\widehat{P}(Z \mid A = a'), P(Z \mid A = a')\Big).$$

This follows from the contractivity of total variation distance under a common Markov kernel, or equivalently under a common pushforward map. Therefore, the deviation from the population-level demographic-parity guarantee is controlled by the estimation error of the reference distribution $P(Z \mid A = a')$. For finite-support $Z$, this error can be controlled by standard empirical distribution theory in terms of the reference-group sample size. For continuous or high-dimensional $Z$, the rate depends on the chosen distribution estimator and structural assumptions, and we leave a systematic finite-sample analysis to future work.

**Computational complexity.** The proposed path-specific counterfactual framework is conceptually agnostic to the ambient dimension: given a specified SCM and a partition of admissible and impermissible paths, the same intervention semantics applies in high-dimensional settings. The main difficulty in such settings lies in specifying or estimating the high-dimensional SCM, rather than in evaluating a given counterfactual query.

Once the SCM and structural equations are specified, the computational cost of a counterfactual query is determined by the subgraph affected by the intervention. Let $A_{\text{aff}}$ denote the set of variables affected by the intervention, and let $E_{\text{aff}}$ be the

edge set of the affected subgraph. Variables outside this affected subgraph do not need to be recomputed. Each affected variable is evaluated once in topological order, so the cost of one counterfactual query is

$$O\left(|E_{\text{aff}}| + \sum_{v \in A_{\text{aff}}} C_v^{\text{pred}}\right),$$

where $C_v^{\text{pred}}$ denotes the cost of evaluating the structural equation or prediction map associated with variable $v$. Thus, the computational cost scales with the size of the affected subgraph rather than the full graph. In practice, the dominant cost typically comes from evaluating the structural or prediction models, while the graph traversal itself is comparatively cheap.

## D. Additional Experimental Details

**Synthetic experimental details.**    The synthetic experiment is conducted with a total sample size of 2000. The data-generating process is described in the main text. The dataset is randomly split into a training set (70%) and a test set (30%), where the training data are used to fit all predictive models and the test data are used to evaluate predictive accuracy.

The variables $W$, $M$, and $Y$ are predicted using $k$-nearest neighbors (KNN) regression models. For the prediction of $W$ and $M$, the number of neighbors is fixed to $k = 35$. For the outcome variable $Y$, the optimal number of neighbors is selected by randomly splitting the training data into an $80\%/20\%$ train–validation split, yielding $k = 14$. This value is then fixed when fitting the final $Y$ prediction model.

Residual-based abduction is performed for the intermediate variables $W$ and $M$, where noise terms are estimated as $W - \hat{W}$ and $M - \hat{M}$, respectively. These estimated noise terms are combined with the structural equations to generate counterfactual realizations of $W$ and $M$ under the proposed path-aware intervention scheme. Counterfactual predictions for the outcome variable $Y$ are obtained by applying the trained predictive model to the counterfactual inputs, yielding predicted probabilities $\hat{p}_Y$, which are converted to binary outcomes using a threshold of $0.5$.

**COMPAS experimental details.**    After applying the filtering criteria described in the main text, the resulting COMPAS dataset contains a total of 6,150 individuals, including 3,696 African American defendants and 2,454 Caucasian defendants. The dataset is randomly split into a training set (75%) and a test set (25%).

The sensitive attribute $A$ encodes race, where African American individuals are assigned $A = 0$ and Caucasian individuals are assigned $A = 1$. The age variable is standardized to have zero mean and unit variance using the sample mean and standard deviation computed from the data. The number of prior offenses exhibits a highly skewed distribution and substantial variability. To improve numerical stability and reduce the influence of extreme values, we apply a logarithmic transformation of the form $\log(1 + \text{priors})$. Gender is encoded as a binary variable, with males assigned a value of 1 and females assigned a value of 0.

The outcome variable $Y$ is predicted using a standard logistic regression model, which estimates the conditional probability $\mathbb{P}(Y = 1 \mid \cdot)$ given the input features. Binary predictions are obtained by thresholding the predicted probabilities at $0.5$, where values greater than $0.5$ are classified as $Y = 1$ and values less than or equal to $0.5$ are classified as $Y = 0$. For interventions on the age variable, we adopt a distributional intervention scheme and generate counterfactual age values by sampling from the conditional distribution $P(\text{age} \mid A = 1)$. For interventions on the gender variable, we use gender $= 1$ as the reference value and construct counterfactual inputs by setting the gender feature to this reference level.

