# OpenReview forum: "The Fairness Hierarchy: A viewpoint from causal inference"
_ICML.cc/2026/Conference — ICML 2026 regular_

### Official Review · Reviewer_w74D · 2026-02-26

**Soundness:** 4
**Presentation:** 3
**Significance:** 3
**Originality:** 3
**Overall Recommendation:** 4
**Confidence:** 3

**Summary:**

This paper defines fairness at the level of causal mechanisms and organizes fairness notions into a hierarchy based on which paths are permissible. They show that under confounding, intervention on multiple sensitive attributes can recover demographic parity.

**Compliance With Llm Reviewing Policy:**

Affirmed.

**Key Questions For Authors:**

Please see limitations.

**Limitations:**

No. See weaknesses for suggestions.

**Strengths And Weaknesses:**

**Strengths**
The paper proposes an interesting intervention and also a clever way to connect causal notions of fairness to demographic parity (via the hierarchy and multi-level intervention). To my knowledge, the results are sound and quite novel.

**Weaknesses**
I only have minor notes.

I think the paper would benefit from adding a section about assumptions and limitations. For example, the “On identifiability” paragraph is an inherent limitation of these SCM and causal fairness-type works. This should be highlighted as a limitation more clearly for readers who might not be as familiar with this literature.

Regarding assumptions, a key point in this paper is that all confounders must be at least partially observed. This is immensely important to highlight from the get-go because this is a strong assumption; in reality there are almost always confounders that are unobserved. There is a reason why most works have mainly dealt with non-confounding (though admittedly also a strong assumption). Discussing why it’s a reasonable assumption, and the implications when they are mildly violated in practice, is important. The authors could even experiment with this in the synthetic experiments, though this idea is not a prerequisite for publication.

On that note, I wasn’t sure what the authors meant by a “learned causal graph” in the abstract, since in theory these results only work relative to a well-specified causal graph; learning these causal relationships is certainly beyond the scope of this paper.

I also think that some of the experiment claims don’t necessarily match with the theory. For example, in both synthetic and COMPAS experiments, the authors say that in line with their theory, predictive performance drops when we enforce stronger path-blocking constraints; this is intuitively true (at least if you fix the model class, as the authors do), but I don’t see that result in the theory – my understanding is that Theorem 3.4 is merely about the possibility of achieving demographic parity, and not the inherent trade-offs one faces with accuracy. If this result is in the paper, please be more explicit with that result if you are going to highlight it in the experiments; otherwise, make sure to be consistent with your claims.

---

> ### Author Rebuttal · Authors · 2026-03-30
>
> We thank the reviewer for the valuable comments and constructive suggestions. Below we address each point and summarize the corresponding revisions.
> >**W1. ...adding a section about assumptions and limitations...**
>
> We are truly grateful to the reviewer for this highly constructive and insightful suggestion. In the revision, we will add a dedicated 'Assumptions and Limitations' section to explicitly clarify three key points: 1) The framework fundamentally relies on a specified structural causal model.  2) Certain fairness levels strongly assume that relevant confounders are at least partially observed.  3) Our current theoretical analysis does not explicitly account for statistical estimation errors. We will also highlight these limitations in the conclusion as critical directions for future work.
>
> >**W2. The strong assumption that all confounders must be at least partially observed, and the implications when it is mildly violated in practice.**
>
> Thank you for this valuable comment. We agree that confounder observability is important for interpreting the scope of our fairness guarantee. In the simple structure $Z \to A$ and $Z \to Y$, if $Z$ is unobserved and the predictor uses only $A$, then omitting $Z$ may affect causal interpretation and predictive accuracy, but does not by itself invalidate the demographic-parity conclusion in our framework, since no observed proxy of $Z$ enters the predictor. However, in the setting considered in our paper, where $Z \to M$ and $M$ lies on the path $A \to Y$, latent information may become partially observable through $M$ and be used for prediction. In this case, exact demographic parity may fail, consistent with Proposition 3.2. To clarify this case, we conducted an additional data analysis under partial observability.
>
> | Metric / Setting | $\hat{Y}^{(0)}$ | $\hat{Y}^{(1)}$ | $\hat{Y}^{(2)}$ | $\hat{Y}^{(3)}$ | $\hat{Y}^{(new)}$ |
> |:----------------:|:---------------:|:---------------:|:---------------:|:---------------:|:-----------------:|
> | Overall accuracy     |   0.923             |       0.833          |        0.757         |        0.838         |        0.683           |
> | Accuracy on $A=0$ |         0.931        |      0.749           |         0.661        |         0.611        |         0.531          |
> | Positive rate    |     0.270            |     0.528            |       0.663          |        0.525         |       0.762            |
>
> **note:** Here “Positive rate” refers to the empirical positive prediction probability for group $A=0$, that is, $\widehat{\mathbb{P}}(\hat{Y}=1 \mid A=0)$. For comparison, the corresponding value for the reference group $A=1$ is 0.663 .
>
> Here, $\hat{Y}^{(\mathrm{new})}$ is obtained by removing $Z$ from the causal graph used in the original experiment while keeping the data-generating mechanism unchanged. In this setting, we still take $A$ as the sensitive attribute and consider the case where the admissible path set is empty. The empirical positive prediction probabilities of the resulting fair output are $0.762$ for group $A=0$ and $0.663$ for group $A=1$, which shows a clear gap between the two groups. This empirical result is consistent with our point above.
>
> >**W3. Confusion over the term "learned causal graph" in the abstract, since the results theoretically rely on a "well-specified causal graph.**
>
> We sincerely thank the reviewer for catching this imprecise wording. We completely agree that learning causal relationships is beyond the scope of this paper, and our framework operates relative to a well-specified causal graph. We will revise the phrase "learned causal graph" to "well-specified causal graph" in a future version of the paper to avoid any confusion.
>
> >**W4. ...some of the experiment claims don’t necessarily match with the theory....**
>
> Thank you for this careful reading and important observation. You are entirely correct: our current theoretical results, including Theorem 3.4, establish the conditions for achieving demographic parity rather than formalizing a general accuracy-fairness trade-off.
> **Our previous phrasing in the experimental section was imprecise.** What we intended to highlight was only an empirical pattern observed in the synthetic and COMPAS experiments: in the specific settings considered there, stronger path-blocking constraints were associated with lower predictive accuracy while producing more similar predictive distributions across groups. However, this is not a theoretical consequence established by our paper, and exceptions may arise within our framework depending on the data-generating mechanism, causal structure, and model class.
> **We will revise the manuscript to ensure strict consistency.** Specifically, we will rephrase the relevant sentences in Section 4 to present this trade-off strictly as an empirical observation demonstrated in our specific experiments, removing any unintended implication that it is a formalized theoretical result of our framework.

---

> > ### Author Rebuttal · Reviewer_w74D · 2026-04-03
> >
> > The authors have addressed my concerns.

---

> > > ### Author Response · Authors · 2026-04-08
> > >
> > > Thank you for your response and for taking the time to review our rebuttal. We sincerely appreciate your constructive comments and suggestions, which have helped improve the paper.

---

### Official Review · Reviewer_4vMw · 2026-03-04

**Soundness:** 3
**Presentation:** 3
**Significance:** 3
**Originality:** 3
**Overall Recommendation:** 4
**Confidence:** 3

**Summary:**

This paper proposes a causal-inference-based framework for fair prediction by specifying fairness directly at the level of counterfactual prediction semantics. Different from previous related work on counterfactual fairness, this paper construct counterfactual predictions by selectively intervening causal paths emanating
from sensitive attributes and this allows a hierarchy of interpretable fairness notions.

**Compliance With Llm Reviewing Policy:**

Affirmed.

**Final Justification:**

Thanks for the author's detailed explanation. However, due to my lack of expertise in the causal inference area, I maintain my initial positive rating after rebuttal.

**Key Questions For Authors:**

- How does the method perform in settings with more variables and more complex causal graphs? How does the computational cost scale?

- Why is there no direct quantitative comparison with existing methods such as Chiappa (2019) and Kusner et al. (2017)?

**Limitations:**

The paper does not have an explicit section about limitations. In the conclusion, the authors briefly mention several limitations about current work regarding efficient estimation procedures for multi-sensitive distributional interventions, integrating with representation learning, and the limited fairness notion of demographic parity. The main limitation may be the strong assumption about the SCM, and there is no robustness analysis addressing this (e.g., if the SCM is not correct, how would the demographic parity guarantee be affected?).

**Strengths And Weaknesses:**

**Strengths**:

- The proposed framework is theoretically solid and well-constructed.
- The definitions, theorems are clearly stated and the proposed framework is consistent and complete, I think this is a meaningful and interesting theoretical contribution to the counterfactual fairness literature.

**Weakness**:

- The main limitation is that the framework relies heavily on a specified SCM (and this may be a common limitation in causal fairness). In practice, obtaining an accurate causal graph is not easy, especially in complicated settings where domain knowledge is limited. This means the proposed method may only work when the causal graph is correct, which is a very strong assumption.

- The second weakness is also related to the strong assumption about the SCM. The method becomes difficult to apply as the number of variables grows. I note that the experiments only involve a small number of variables (including synthetic data), so it remains unclear how well this framework scales to more complex and high-dimensional settings that are common in practice.

---

> ### Author Rebuttal · Authors · 2026-03-30
>
> Thank you for your insightful comments. Below we address your concerns and summarize the revisions made in response to them.
> >**W1. ...the framework relies heavily on a specified SCM...**
>
> Regarding the discussion of the need for a specified SCM, we refer the reviewer to our response to Reviewer zdVf(W1) for details. We additionally provide a numerical analysis under graph misspecification.
>
> Specifically, we revisited the synthetic setting in Section 4.1, kept the data-generating mechanism unchanged, and trained the prediction model under a misspecified SCM obtained by adding the edge $Z \to W$. We then applied the strategy corresponding to $\hat{Y}^{(2)}$, where $(A,Z)$ are treated as the sensitive attributes and the admissible path set is empty. The resulting empirical results are reported under $\hat{Y}^{(\mathrm{new})}$.
> | Metric / Setting | $\hat{Y}^{(0)}$ | $\hat{Y}^{(1)}$ | $\hat{Y}^{(2)}$ | $\hat{Y}^{(3)}$ | $\hat{Y}^{(\mathrm{new})}$ |
> |:----------------:|:---------------:|:---------------:|:---------------:|:---------------:|:--------------------------:|
> | Overall accuracy | 0.923 | 0.833 | 0.757 | 0.838 | 0.747 |
> | Accuracy on $A=0$ | 0.931 | 0.749 | 0.661 | 0.611 | 0.620 |
> | $\widehat{\mathbb E}[\hat{Y}\mid A=0]$ | 0.270 | 0.528 | 0.663 | 0.525 | 0.654 |
> | $\widehat{\mathbb E}[\hat{Y}\mid A=1]$ | 0.663 | 0.663 | 0.663 | 0.663 | 0.663 |
>
> **note:** $\widehat{\mathbb E}[\hat{Y}\mid A=a]$ denotes the empirical conditional expectation of the prediction output within group $A=a$.
>
> Under this mild misspecification, $\hat{Y}^{(\mathrm{new})}$ remains close to $\hat{Y}^{(2)}$ in both accuracy and group-wise output. For $\hat{Y}^{(\mathrm{new})}$, $\widehat{\mathbb E}[\hat{Y}\mid A=0]=0.654$ and $\widehat{\mathbb E}[\hat{Y}\mid A=1]=0.663$, showing no obvious group gap. This suggests that the empirical behavior remains relatively stable under mild SCM misspecification.
> >**W2-Q1. Scalability to more complex and high-dimensional settings and how the computational cost scales as the number of variables grows.**
>
> We thank the reviewer for this insightful question. Our path-specific counterfactual framework is conceptually agnostic to dimensionality: given an SCM and a partition of admissible and impermissible paths, the same formulation applies in high-dimensional settings. The main challenge lies not in the fairness definition itself, but in estimating high-dimensional SCMs and specifying complex causal graphs.
>
> From a computational perspective, let $A_{\mathrm{aff}}$ denote the variables affected by the intervention. Once the SCM and structural equations are specified, only variables in $A_{\mathrm{aff}}$ need to be recomputed, each exactly once in topological order. Thus, one counterfactual query costs $O\left(|E_{\mathrm{aff}}|+\sum_{v\in A_{\mathrm{aff}}} C_v^{\mathrm{pred}}\right)$, where $E_{\mathrm{aff}}$ is the edge set of the affected subgraph and $C_v^{\mathrm{pred}}$ is the cost of evaluating variable $v$. In practice, the main cost comes from prediction rather than graph traversal. This discussion will be added to the revised manuscript.
>
> >**Q2.Why is there no direct quantitative comparison with existing methods such as Chiappa (2019) and Kusner et al. (2017)?**
>
> We thank the reviewer for raising this question. In our hierarchy, the setting of Chiappa (2019) corresponds to the case of a single sensitive attribute. Kusner et al. (2017) introduced individual counterfactual fairness, which, as discussed in Appendix B, is recovered in our framework when there is a single sensitive attribute and the admissible path set is empty, corresponding to $\hat{Y}^{(1)}$ in our experiments. To further address this point, we additionally implemented an experiment motivated by Kusner et al. (2017), using residuals of descendants of $A$ together with non-descendants as predictor inputs, and report the results below. We will clarify these relations and comparisons in the revised manuscript.
>
> | Metric / Setting | $\hat{Y}^{(0)}$ | $\hat{Y}^{(1)}$ | $\hat{Y}^{(2)}$ | $\hat{Y}^{(3)}$ | $\hat{Y}^{(\mathrm{new})}$ |
> |:----------------:|:---------------:|:---------------:|:---------------:|:---------------:|:--------------------------:|
> | Overall accuracy | 0.923 | 0.833 | 0.757 | 0.838 | 0.840 |
> | Accuracy on $A=0$ | 0.931 | 0.749 | 0.661 | 0.611 | 0.846 |
> | $\widehat{\mathbb E}[\hat{Y}\mid A=0]$ | 0.270 | 0.528 | 0.663 | 0.525 | 0.379 |
> | $\widehat{\mathbb E}[\hat{Y}\mid A=1]$ | 0.663 | 0.663 | 0.663 | 0.663 | 0.600 |
>
> **note:** $\widehat{\mathbb E}[\hat{Y}\mid A=a]$ denotes the empirical conditional expectation of the prediction output within group $A=a$.
>
> For the Kusner-style method $\hat{Y}^{(\mathrm{new})}$, we have $\widehat{\mathbb{E}}[\hat{Y}\mid A=0]=0.379$ and $\widehat{\mathbb{E}}[\hat{Y}\mid A=1]=0.600$, indicating a clear gap between the two groups. This suggests that, in our synthetic setting, the Kusner-style method still shows substantial group-level disparity.

---

> > ### Author Rebuttal · Reviewer_4vMw · 2026-04-03
> >
> > Thanks for the detailed explanations. I have no remaining questions.

---

> > > ### Author Response · Authors · 2026-04-08
> > >
> > > Thank you for your response and for taking our rebuttal into consideration. We sincerely appreciate your valuable comments and suggestions, which are very helpful for improving the paper.

---

### Official Review · Reviewer_zdVf · 2026-03-08

**Soundness:** 2
**Presentation:** 3
**Significance:** 2
**Originality:** 3
**Overall Recommendation:** 4
**Confidence:** 4

**Summary:**

The authors propose a path specific definition of counterfactual fairness for prediction that picks a causal graph/SCM, labels admissible vs impermissible causal pathways from sensitive attributes to outcomes, then keeps sensitive information only along admissible paths and setting it to a reference/baseline along impermissible paths. They introduce a multi-intervention formulation of counterfactual prediction that explicitly accounts for confounding structures and show that a specific fairness level of their proposed hierarchy satisfies demographic parity.

**Compliance With Llm Reviewing Policy:**

Affirmed.

**Final Justification:**

I'd like to thank the authors for taking the efforts to address my concerns. I have increased my score accordingly.

**Key Questions For Authors:**

1. could the authors comment on how sensitive are the results to misspecification of the causal graph / scm?
2. what's the sample complexity of estimating the distribution intervention $Z\star$ and how does the estimation error propagate to the final fairness guarantee?

**Limitations:**

yes

**Strengths And Weaknesses:**

Soundness: The overall framework and architecture aligns with the literature. I understand the authors have acknowledged that the path specific effects may not be fully identifiable when there is unobserved confounding, but I am still concerned with the practical usefulness / robustness of this framework when in the real world our knowledge of DAG or SCM may not be fully correct. Theorem 3.4 relies on drawing the distribution of $Z^\star$ and knowing the conditional distribution of the confounders, so how does the estimation error propagate eventually?

Presentation: The running example is helpful for the readers to understand the framework and the progression from single sensitive to multi sensitive to distributional results is also paced very well. However, there seems to be a few typos / notation inconsistency like around definition 2.5, there is reference to definition 2.3 but there is no definition 2.3.

Significance: The paper clearly is working on an important and interesting question of having a more flexible path specific counterfactual framework for fair prediction. The definitions all make sense but seem to stay merely as conceptually appealing principles whose practical impact is not yet fully justified in this work.

Originality: Allowing intervention targets to vary by path is a generalization / extension of prior work and makes a lot of sense.

---

> ### Author Rebuttal · Authors · 2026-03-30
>
> Thank you for your valuable comments. Our responses below address the concerns raised in your review.
>
> >**W1. Robustness and sensitivity of the framework to potential DAG or SCM misspecification.**
>
> We thank the reviewer for raising this important point. We agree that a specified DAG/SCM is a basic premise of our framework. More generally, this is not unique to our approach, but is required by any method that aims to reason about causal-level fairness rather than only statistical parity. We address robustness and sensitivity to misspecification from two practical perspectives：
>
> - **Robustness through Transparency:** If the assumed graph is misspecified, any causal method may deviate from the true mechanism. The advantage of our framework is that it makes the admissible and impermissible paths explicit, so the fairness logic can be audited and revised by domain experts instead of being hidden in a black-box procedure.
>
> - **Sensitivity to graph uncertainty:** In practice, some parts of the causal graph may remain uncertain. Still, our framework can provide meaningful guidance on sufficiently specified sensitive pathways, with uncertainty limited to unresolved parts. We will clarify this limitation and the importance of sensitivity analysis in the revision. **Due to space limitations, our new empirical analysis on SCM misspecification is provided in the responses to Reviewer 4vMw (W1) and Reviewer w74D (W2).**
>
> >**W2. Minor typos and notation inconsistencies (e.g., missing Definition 2.3 reference)**
>
> We thank the reviewer for the careful reading and for catching this presentation issue. The reference to `Definition 2.3` is a cross-reference typo; it should instead refer to `Definition 2.1`. We will correct this in the revision and carefully proofread the paper for similar notation inconsistencies.
>
> >**W3. Further justification for the practical impact and empirical utility of the proposed framework.**
>
> We appreciate the opportunity to clarify this. Our framework's empirical utility lies in translating abstract causal fairness into an actionable tool for practitioners. We highlight two key practical advantages over existing methods:
> - **Surgical Interventions for Complex Confounding:** Real-world discrimination is rarely single-variable. As shown in our COMPAS experiment (Sec 4.2), the sensitive attribute (Race) is confounded by demographics (Age/Gender). Our multi-sensitive framework practically enables intervening at different locations along distinct paths, effectively resolving complex confounding that standard single-variable methods cannot handle.
> - **Plug-and-Play Model Agnosticism:** Because the framework operates purely at the counterfactual prediction level ($Y^{fair}$), it can be seamlessly wrapped around any optimized, pre-trained base predictor without needing to alter internal training dynamics, loss functions, or architectures.
>
> We will highlight these practical advantages in the revised manuscript to better ground our theoretical contributions.
>
> >**Q1: what's the sample complexity of estimating the distribution intervention $Z^\star$ and how does the estimation error propagate to the final fairness guarantee?**
>
> We first clarify that Theorem 3.4 is a population-level result without statistical estimation error, characterizing the link between our framework and demographic parity in an idealized setting.
>
> When $A$ is binary, estimating $P(Z \mid A=a')$ reduces to estimating the distribution of $Z$ in one subgroup. A natural plug-in estimator is the empirical conditional distribution based on samples with $A=a'$, for which standard consistency and concentration results apply.
>
> A key point is that the exact demographic-parity conclusion in `Theorem 3.4` relies on the identity in distribution $Z^\star \sim P(Z \mid A=a')$. If this reference distribution is replaced by an estimate $\widehat P(Z \mid A=a')$, then exact demographic parity is generally replaced by an approximate version.
>
> To quantify this deviation, let $d_{\mathrm{TV}}(P,Q) := \sup_B |P(B)-Q(B)|$ denote the total variation distance. In the proof of `Theorem 3.4`, the fair outcome can be represented as $Y^{\mathrm{fair}}=\phi(a',Z,X,U)$ for the reference group and $\widehat Y^{\mathrm{fair}}=\phi(a',\widehat Z^\star,X,U)$ for the plug-in intervention group, where $\widehat Z^\star \sim \widehat P(Z \mid A=a')$. Therefore, the induced fairness gap can be bounded as
> $$
> d_{\mathrm{TV}}\left(
> \mathcal{L}(\widehat Y^{\mathrm{fair}} \mid A=a),\;
> \mathcal{L}(Y^{\mathrm{fair}} \mid A=a')
> \right)
> \le
> d_{\mathrm{TV}}\left(
> \widehat P(Z \mid A=a'),\;
> P(Z \mid A=a')
> \right).
> $$
> This is because the fair outcome distribution is the pushforward of the intervention distribution under the same SCM/prediction map, and total variation is contractive under a common Markov kernel. Thus, the deviation from exact demographic parity is controlled by the estimation error of $P(Z\mid A=a')$. We will add this discussion to the revised manuscript.

---

> > ### Author Rebuttal · Reviewer_zdVf · 2026-04-03
> >
> > Thanks for the detailed response.

---

> > > ### Author Response · Authors · 2026-04-08
> > >
> > > Thank you for your response and for considering our rebuttal. We sincerely appreciate your helpful comments and suggestions, which have been valuable for improving the paper.

---

### Decision · Program_Chairs · 2026-04-30

**Decision:**

Accept (regular)

**Comment:**

The reviewers thought that the paper was clearly written, well motivated and found the proposed framework to be theoretically solid and well-constructed. Regarding concerns:
(1) The assumption of a known DAG. The authors clarified that the method can still provide meaningful conclusions in settings where the graph is partially known for parts of the graph that are correctly specified. They also provided an empirical sensitivity analysis for violations of this assumption
(2) Practicality of the proposed approach: the authors clarified how it could be used in practical settings through surgical interventions for complex confounding and by being a flexible approach that is model agnostic.
(3) Scalability as the number of variables grows: authors clarified that scaling depends on the number of variables affected by the intervention not the number of variables in the DAG